# Interpretive Trends and the Conceptual Construction of the Daodejing's Dao in Russian Sinology: A Historical Overview

## Hongyan Zhang and Jing Luo *

School of Foreign Languages and Literature, Wuhan University, Wuhan 430072, China
* Correspondence: luojing1987@whu.edu.cn

**Abstract:** "Dao" is not only a core concept in the *Daodejing*, but is also an important keyword in Chinese classical philosophy. It encompasses the origin of all things in the universe, the laws of nature, and the laws of dealing with the world. A historical review of Russian sinologists' interpretation and conceptualization of "Dao" reveals the differences in philosophical understanding and translation strategies of those sinologists, and reflects the translation loss and compensation of Chinese classical philosophical keywords in the process of foreign dissemination. During the Imperial Russian Period, researchers regarded "Dao" as the revitalization of religious theology. During the Soviet Period, the aim of the Soviet researchers was to find the struggle between materialism and idealism in "Dao". In the Post-Soviet Period, researchers gradually threw off the shackles of ideology, and began to conduct more diversified and multi-level research on "Dao" and the *Daodejing*. This article aims to discuss the research and translation of the *Daodejing* in Russia, paying particular attention to the dissemination and reception of "Dao" in Russia. It also endeavors to explore the interpretive trends of "Dao" in Russia and highlight the dissemination and understanding of Laozi thought in Russia.

**Keywords:** Dao; the *Daodejing*; Russia; sinology; materialism

## 1. Introduction

The *Daodejing* is the core philosophical classic of Daoism and is an important part of world civilization. The *Daodejing* and Daoist thought first came to Russia in the first half of the 19th century, in which the Russian Orthodox Mission played a very important role. From 1715 to 1956, Russia sent 20 Orthodox Missions to China, which became the cradle of Russian sinology[1]. Among them, a large number of sinologists emerged, making outstanding contributions to the dissemination of Chinese culture and the promotion of Confucianism and Daoism. The *Daodejing* is a worldwide phenomenon, and it has been translated into most of the global languages that are in use today. The research on Daoism and Laozi in Russia started relatively late, but the *Daodejing* is the most frequently translated Chinese classic in Russia, second only to the *Bible* among the world's famous classics. According to Dr. Misha Tadd, there are 42 Russian translations of the *Daodejing* (Tadd 2019, p. 105). These translations have outstanding academic value, and they represent a microcosm of Russian translation and research on Chinese Daoist philosophy.

With "Dao" at its core, the *Daodejing* constructs a rich philosophical system involving figures ranging from emperors who reign the world to hermits who value self-cultivation. The entire Daoist philosophical system develops with the "Dao", a concept formed by Laozi. The understanding and reception of "Dao" in Russia have been deeply influenced by social ideology, and are closely related to the historical development of Russian sinology. With the changes in time and space, the interpretation and conceptualization of "Dao" in Russia can be divided into three stages. The first stage was the Imperial Russia Period (from early 19th century to the beginning of the 20th century). Scholars in this stage compared the interpretation of "Dao" with Christianity, and their research was illusive and had a touch of Eurocentrism. The second stage was the Soviet Period (from the beginning of the

20th century to the end of the 20th century), during which academic research was usually measured by political standards. The study and understanding of "Dao" also became one of the main positions of the struggle between materialism and idealism. During this period, materialists regarded "Dao" as a powerful weapon against religious thought, whereas idealists mystified "Dao" to disarm materialism. The third stage was the Post-Soviet Period (from the end of the 20th century to the present), and the researchers in this period tended to be more objective and reasonable in their understanding and attitude toward "Dao" and began to adopt scientific methods to conduct multifaceted, in-depth, and specific research on Daoism.

This article adopted the keyword research method proposed by R. Williams, which opened up a new way of using historical semantics for social and cultural research. According to Williams, when conducting an in-depth interpretation of a keyword in a certain field, not only do we need to emphasize the historical origins and developments of the semantics, but more importantly, we need to pay attention to the radical change, discontinuity, and conflict in different interpretations of the keyword. In this article, we took "Dao", a keyword and core concept in the *Daodejing*, as the research object, and endeavored to investigate the understanding and interpretation of "Dao" in Russia, and explore the interpretive trends and reception of the *Daodejing* in Russia.

## 2. The Imperial Russia Period—The Revitalization of Religious Theology

The Imperial Russia Period was an important time for Russia to realize foreign expansion, and great importance was attached to the study of the culture and economy of neighboring countries. At that time, Russian sinologists were mainly members of the Russian Orthodox Missions, and their academic research was conducted under official instructions and monitoring. Under the guise of religious propaganda, they were essentially studying China's economy, politics, and culture while collecting information and intelligence about China for the Russian government. Back then, Western missionaries generally believed that the *Daodejing* implied the teachings of the *Bible*, and the image of "Dao" was basically manifested through the metaphors of "God" or "Creator", which was also because early translators adopted a target-culture-oriented translation strategy[2]. Restricted by the social environment, Russian sinologists in this period had strong historical and religious limitations when studying and approaching the concept of "Dao", which was always associated with keywords such as "divine", "God", "mysticism", "prophet", etc.

In 1818, the Russian government explicitly instructed the Tenth Orthodox Mission to study the Chinese religion. Count Speranskij Mikhail Mikhajlovich (1772–1839) wrote the "Summary of Instructions and Questions Offered by the Academy of Sciences for Young Travelers on Mission to China" (Nachertanie instruktsij i voprosov, predlagaemykh Akademiej nauk v pol'zu i upotreblenie molodym puteshestvennikam, naznachennym i otpravlyaemym pri dukhovnoj missii v Kitaj), which stipulated that once the priest had enough knowledge of the Chinese language, they should start to study Buddhism and Daoism, translate books that help explain the teachings of these two religions, and prepare materials and arguments needed to rebut the two religions (Skachkov 1977, p. 128). It was under this circumstance that the monk priest of the Tenth Orthodox Mission, Archimandrite Daniil (Sivillov Dmitrij Petrovich) (1798–1871), started the translation of the Daodejing. During the translation process, Sivillov failed to find the so-called "rebuttal evidence", instead, he was amazed and overwhelmed by the philosophical wisdom in Laozi's thoughts and believed that the Daodejing contained wisdom that the Analects did not possess. Sivillov associated "Dao" with immortality of the soul, the immortality of life and karma, giving "Dao" a mystic touch. He wrote: "The 'Dao' is the creator of the universe, the wisdom, the rules, the judge of the law, the spirit that rules everything like God" (Khokhlov 2014, p. 493). In 1826, he completed the first manuscript[3] of the Russian translation of the Daodejing, which could be regarded as the beginning of Russian research on the Daodejing. Regrettably, Sivillov's translation was not approved for publication until 1915 when "The Unpublished Daodejing in the Daniil (Sivillov) Archives" (Neopublikovannyj per. Dao-

deh-tszina arkhimandrita. Daniila (Sivillova)) was published in the Proceedings of the Odessa Bibliographic Society (Izvestiya Odesskogo bibliograficheskogo obshhestva) [only includes Chapter 1–Chapter 46].

During this period of time, a lot of research was conducted on Laozi by Western scholars. For example, Jean-Pierre Guillaume Pauthier (1801–1873), a French Orientalist, published Le Tao-Te-King in 1838. In 1842, the famous French sinologist Stanislas Julien produced the first complete translation of the Daodejing, followed by Victor Von Strauss, who translated it into German in 1870. Later, an English version of the Daodejing, undertaken by a Scottish sinologist James Legge (1815–1897), came out in 1891. Then, in 1898 and 1903, Paul Carus (1852–1919) and Isaac Heysinger published another two versions of English translation, respectively. In 1842, the 11th issue of Son of the Motherland (Syn Otechestva) published an anonymous article introducing Laozi, which was regarded as the first published translation and research work on the Daodejing in Russia. The author quoted and translated the French translation of the Daodejing by Stanislas Julien (1797–1873). The research on the anonymous authorship of this article was quite controversial. Russian bibliography experts generally believed that the author of the article was Bichurin Nikita Yakovlevich (1777–1853), whereas Khokhlov Aleksandr Nikolaevich (1929–2015) thought the author was Senkovskij Osip Ivanovich (1800–1858), as pointed out in the report "Who is the Author of the Article 'Laozi and His Teachings' in the Journal Son of the Motherland in 1842" (Kto avtor stat'i "Lao-tszy i ego uchenie" v zhurnale Syn otechestva 1842 g.) collected in the 19th All-Russian Conference of Philosophy and Modern Civilization of the East Asian Region. Be it Bichurin or Senkovskij, the publication of this article had groundbreaking significance for the spread of Daoism in Russia, and it proved, to some extent, that the early formation and development of Russian sinology was deeply influenced by European sinology, as most of the early translations of Chinese classics were translated from English and French translations.

In the history of Russian sinology, the representative figure of the second half of the 19th century was Vasil'ev Vasilij Pavlovich (1818–1900). Back then, the fortress of Russian sinology research gradually shifted from the Russian Orthodox Mission to universities and research institutes, which was marked by the book *Religions of the East: Confucianism, Buddhism and Daoism* (Religii Vostoka: Konfutsianstvo, buddizm i daosizm) by Vasil'ev. In this book, Vasil'ev praised, analyzed, and translated the Daodejing. He spoke highly of it, believing that "Laozi's language is very unique, and the ideas expounded in the book are more profound compared with that of the Analects, the grammar more accurate compared with that of Zhuangzi and Mengzi, it is thus simpler and easier to understand" (Vasil'ev 1873, p. 76). However, Vasil'ev did not remove the utopian mysticism in his understanding of the ideological connotation of the Daodejing; he even speculated that Laozi had been to the West and that the Daodejing was completed on the way to the West. He compared the Daodejing to the *Bible*, believing that the ideas reflected by the three Chinese characters of Dao道, de, and jing经 were very close to the concept of God. He also divided Daoism into "Daoist philosophy" and "Daoist religion" from the perspective of positivism. It was his belief that Russia and Europe had the mission to enlighten the East, as Daoist thought was conservative and backward. It is not hard to find evidence of Eurocentric tendency in his viewpoint. During Vasil'ev's time, the research on Daoism in Russia was not sufficient. From Vasiliev's two books, *Religions of the East: Confucianism, Buddhism and Daoism* (Religii Vostoka: Konfutsianstvo, buddizm i daosizm) and *Outline of the History of Chinese Literature* (Ocherk istorii kitajskoj literatury), we can see that the research on Confucianism accounted for more than half of the works and the research on Buddhism was more profound, but less attention was paid to Daoism. This is also supported by the arguments in the book *On the Science of the East* (Nauka o Vostoke), in which Alekseev Vasilij Mikhajlovich (1881–1951) divided the Russian sinology education of the second half of the 19th century and the first half of the 20th century into two periods, with the Chinese Revolution of 1911 and the Russian Revolution of 1917 as the watershed. Alekseev wrote in the book: "In the second period, one of the biggest breakthroughs in Sinology education was that, the Daodejing

was incorporated into the Russian syllabus, becoming part of the Chinese classics. Whereas in the first stage (i.e., before the revolution), students only studied Confucian literature, which was not enough, we have to recognize the importance of Laozi's teaching and read intensively about it. "(Alekseev 1982, p. 174)".

Although having studied under Vasil'ev, the famous sinologist Georgievskij Sergej Mikhajlovich (1851–1893) had a different view from his teacher on the value of traditional Chinese culture. He opposed the Eurocentric tendency in Chinese philosophical research and extensively used atypical positivism expressions such as "world material" (mirovaya materiya) and "world spirit" (mirovoj dukh) to analyze the traditional Chinese philosophical thought including the interpretation of "Dao". He pointed out that "As the material and concept that came into existence at the very beginning, Dao is the only eternal and unchanging matter in the myriad things in nature...Dao is both the world's material, the world's power, and the world's spirit" (Georgievskij 1885, pp. 299–300). With the deepening of his research, Georgievskij tried to explain Daoism from the perspective of absolute materialism and claimed that material develops according to its internal rules, rather than the interference of some concepts. He interpreted "Dao" as the law of nature, "Virtue appears when human beings follow the law of nature (i.e., 'Dao')" (Georgievskij 1892, p. 112).

The encounter between the famous writer Tolstoj Lev Nikolaevich (1828–1910) and Laozi took place at the right time. In 1877, Tolstoj completed the work *Anna Karenina*, and then found himself in a spiritual and existential crisis, so resorted to traditional Chinese thoughts to find a way out of his predicament. Through Strakhov Nikolaj Nikolaevich (1828–1896), Tolstoj obtained the Daodejing translated by French sinologist Stanislas Julien. Based on this translation, he selected and translated the chapters that he thought were worth translating. From Tolstoj's translation and interpretation of "Dao", we can see the continuous evolution of his understanding of Daoism. When he came into contact with the French version of the Daodejing, he used the word "God" (bog) to interpret "Dao". In his view, "Dao" was the symbol of God as well as the way to God. He believed that Laozi's theory was, in essence, similar to Christianity. In his book *A Book about Path and Truth Written by the Chinese Sage Laozi* (Kniga Puti i Istiny, napisannaya itajskim mudretsom Laotsy) (1884), Tolstoj wrote: "Dao is obtained through the temperance of all personal and carnal things...The essence of both is the spirit and divinity that form the basis of human life, manifested in ascetic ways. Therefore, for human beings to become a blessing instead of a trouble, one should learn to live not for material desires but for the spirit, which is exactly what Laozi taught" (Tolstoj 1956, pp. 350–51). The "God" he meant was not the "God" in the sense of the church, but the beginning of the human soul, and the subjective love and objective happiness. This is also the cornerstone of Tolstoyism[4]. Later, Tolstoj further developed his own interpretation of "Dao". In his article "Non-action 無爲" (Nedelanie) (1893), he translated "Dao" as "path, virtue, truth" (put', dobrodetel', istina). He believed that the acquisition of "Dao" must be achieved through doing nothing that goes against nature, and based on this, he proposed "non-resistance to evil by violence" and "moral self-improvement". Later, in the process of proofreading the Russian translation of the Daodejing by Konissi Masutarō 小西増太郎 or Konissi Daniil Petrovich (1862–1940), Tolstoj developed a new understanding of Laozi's thought. In the *Anthology of Daily Thoughts of the Wise* (Mysli mudrykh lyudej na kazhdyj den') (1903), "Dao" was translated into reason (razum). In 1906, Tolstoj made further changes and reflections on "Dao" in his open letter "A Letter to a Chinese" (Pis'mo k kitajtsu) to Gu Hongming 辜鸿铭, in which he perceived "Dao" as "freedom" (svoboda). This freedom did not mean freedom from shackles, but rather what one acquired after knowing the law of nature. In his subsequent research, he synthesized his previous understanding and interpretation, believing that "the law of 'Dao' means rational life is the only way to be cherished, and 'Dao' is the necessary and supreme law of heaven or God" (Tolstoj 1935, pp. 295–98). It can be seen that Tolstoj's understanding of "Dao" expanded and changed over the years, because he translated indirectly through the English, French, and German versions to Russian, in the process of which, he added

his own understanding. Laozi's thinking, Western philosophical research on Laozi, and Tolstoj's own world view were all reflected in Tolstoj's works.

In 1893, Tolstoj's follower Konissi published "The Philosophy of Laozi" (Filosofiya Laosi) in the Issues of Philosophy and Psychology (Voprosy filosofii i psikhologii), and later published the first complete translation of the Russian version of the Daodejing in 1894 with the support of Tolstoj. He argued that Laozi's philosophy was an idealist philosophy and wrote: "For Laozi, all concrete substances belong to the category of existence and are constantly changing; Laozi proved that there is an eternal, unchanging category that includes all existence, namely 'Dao.' 'Dao' is the purest spirit, with internal unity and inseparability" (Konissi 1893, p. 42).

Konissi received a European-style education, initially following the Orthodox Church, and then Tolstoyism. Therefore, his research on Daoism was carried out under the framework of Western philosophy rather than Chinese philosophy. He compared Laozi's thought with that of Western philosophers such as Heraclitus (about 544–483 BC), Plato (427–347 BC) and the Eleatic School, believing that the concept closest to "Dao" in Western philosophy was "nous" proposed by Anaxagoras (500–428 BC). Konissi was convinced that "Similar to Greek philosophy, Laozi's metaphysical system is also a systematic and complete exposition on the highest existence. The difference lies in the fact that Laozi's thought is the product of individual wisdom, whereas Greek philosophy is the product of joint effort of many scholars" (Konissi 1894, pp. 386–88). It is worth mentioning that Konissi's version of the Daodejing was the only complete Russian translation until 1950, thus is a valuable asset in the Russian study of Laozi with strong research significance.

Contrary to Tolstoj and Konissi, who commended and respected Daoism, Solov'yov Vladimir Sergeevich (1853–1900), who played a significant role in the development of Russian philosophy and Orthodox theology, believed that both Confucianism and Daoism refuse to create, with the former holding the ancestors in high esteem and the latter advocating non-action, therefore laying the foundation of Chinese conservatism. Influenced by Christian-centralism and Eurocentrism, Solov'yov regarded China as an alien and dangerous force. He argued that the essence of Chinese thought was that the past had unconditional dominance over the present, and Laozi's theory was a full testimony of that. "The theory of Laozi is a ridiculous philosophy, just like other Chinese people, Laozi intends to seek the absolute origin of the world from the past, a past that is unconditional and exists above all matter, this negative force is the so-called 'Dao.' All things in the world originate from it and eventually return to it, that is to say, 'Dao' is the path that all things follow in common, yet there is no clear definition for this absolute origin of all things, because it is simply indescribable and unspeakable" (Solov'yov 1996, p. 119). In his view, the essence of "Dao" is "non-action", which is the embodiment of obscurantism and denial of life, knowledge, and progress. It is exactly with the analysis of Laozi's "Dao" that he explained the philosophical origins of "conservatism" and "traditionalism" in Chinese culture. As a philosopher trained in the Western philosophical system, Solov'yov approached and examined a mysterious culture of the other, that is different from the West, from a Western point of view. Notwithstanding differences, his interpretation of "Dao" rested on the patterns of 19th century European Orientalism[5], emphasizing East–West distinction and Western superiority.

During this period, Russia not only obtained information on sinology indirectly from Western Europe, but also focused on developing its own sinology. Russian sinologists tried to make comparisons and interpretations of "Dao" by incorporating it into the research field of comparative philosophy. Granovskij Timofej Nikolaevich (1813–1855), a Professor at Moscow University, believed that Laozi was obviously familiar with the speculative philosophy of Hinduism, since he preached benevolence and opposed excessive material enjoyment. Laozi also deemed that everyone should lead a life that is different from a worldly life, and he was a believer that the eternal and absolute reason (the "Dao") was the origin of all existence (Granovskij 1990, p. 609). According to Granovskij's world view, the mechanical view of nature coexists with the organic view of nature. He believed that

"the eternal and absolute reason" was the "Dao", which was very consistent with Hegel's system of absolute idealism, but it did not contradict Kant's positivist theory, and this was the first attempt by a Russian researcher to use the idealism to interpret Daoism. In his book *Religion in China* (Religiya Kitaya), Glagolev Sergej Sergeevich (1865–1937), a Professor at the Moscow Orthodox Theological Seminary, compared and analyzed "Dao", a main concept in the quest of cultural generation, operation mechanism and law, with "Logos", he believed that "'Dao' is the path, the follower of this path as well as the path followed by all things. 'Dao' is not created, because 'Dao' itself is eternal. 'Dao' is everything and nothing, a cause and an effect" (Glagolev 1901, p. 33). Before the October Revolution, Russian sinologists compared the Daodejing with religion when translating and interpreting the Daodejing. Their interpretation of "Dao" was mysterious, conservative, and religious, with strong historical and religious limitations as well as Eurocentrism and Slavism[6] tendencies.

### 3. The Soviet Period—Materialist Understanding Due to Ideology

After the October Revolution in Russia, the establishment of Soviet Marxism–Leninism and the Chinese Revolution had a significant impact on the research objects and methods of Soviet sinology. Unlike the previous Imperial Russian sinology, the sinology research during the Soviet Period became inextricably intertwined with political reality. The research and understanding of "Dao" were freed from the religious vision of divinity and prophets, and a unique academic research school was formed with new perspectives and research methods. Three stages can be identified during this period. The first stage was from after the October Revolution in 1917 to 1935, which was a period of "pure academic" research with few political involvements; the second stage spanned roughly from 1935 to 1966, in which the politically motivated "New Daoism" prevailed. During this period, the Lao-Zhuang doctrine was interpreted as materialistic and progressive. In the third stage between 1966 to 1983, the materialistic interpretation of "Dao" was questioned and challenged, and there was a dispute between the materialist and idealist perspectives, with the former still taking the upper hand.

In the early days when the Soviet Union was founded, purely academic discussions on Daoism were the mainstream. Sinologists who conducted traditional academic research without the influence of social and political realities could be categorized as "old-school" Daoist researchers, as represented by Alekseev, whose understanding of Daoism had traces of idealism and mysticism. He believed that "'Dao' is a rule, an absolute truth beyond human understanding, and it is eternal. Human is the third element after Heaven and Earth, and the 'Dao' lies in the heart of Human" (Alekseev 1978, p. 49).

Alekseev's disciple Petrov Apollon Aleksandrovich (1907–1949) can be regarded as a trailblazer of the "new-school" of Daoism research. In the early stage of his research, Petrov was influenced by his teacher and asserted that Daoist philosophy followed the system of objective idealism. When refuting the views of Solov'yov, a pre-revolutionary scholar, Petrov wrote: "Instead of being a negative force, as the author (i.e., Solov'yov) thinks, 'Dao' represents an absolutely positive force. Under the framework of idealism, 'Dao' is interpreted as a kind of uncertain potential in existence, and at the same time, it is also the absolute and only actual existence" (Petrov 1935, p. 10). With the publication of Stalin's "Dialectical Materialism and Historical Materialism" (*O dialekticheskom i istoricheskom materializme*) in 1938 and the in-depth research of Petrov, Petrov gradually became a pioneer who drew on Marxism to explore the history of Chinese philosophy. Previously, Petrov broke away from the specific historical context and conducted logical analysis to discuss the abstract reality of Daoist theory. In the book *Introduction to Chinese Philosophy* (*Ocherk filosofii Kitaya*) published in 1940, scholars began to turn to the method of historical comparison, seeing early Daoism as a stage of philosophical development, explaining the logic and trend of its development, and successfully finding evidence of materialism (even rationalism) in Daoism. Petrov pointed out:

Accurately speaking, there is no widely accepted Russian translation and interpretation for the concept of "Dao". Existing translations and interpretations (logos, world-way,

god, world-cause, spiritual power, pure transcendental being, etc.) fail to express the true essence of "Dao", making it a principle that exists only in ideas, without taking into account its basic definition. Based on this account, two points must be noted: Dao operates in accordance with the natural conditions of all things. Dao comes into existence before Heaven and Earth. It suggests that "Dao" may have a materialistic connotation in its ancient understanding, and "Dao" may be a material existence that follows the law of natural development, and it also covers the law of evolution of such existence. (Petrov 1940, pp. 251–52).

Building on Petrov's research, the overseas Chinese scholar Yang Xingshun 楊興順 (1904–1987) became the most influential representative of "new-school" Daoism researchers in the study of Laozi and Zhuangzi in Russia. If Petrov was trying to find a materialist motive in Laozi, Yang Xingshun made Laozi a staunch materialist. In 1947, Zhdanov Andrej Aleksandrovich (1896–1948), a member of the Political Bureau of the CPSU Central Committee in charge of ideological and political work in the Soviet Union at that time, personally chaired a seminar on the *History of Western European Philosophy* (*Istoriya zapadnoevropejskoj filosofii*) written by Aleksandrov Georgij Fedorovich (1908–1961). He wrote: "The history of philosophy is mainly the history of materialism development, and idealism philosophy is only allowed to appear in the history of philosophy as the object of criticism for materialists" (Zhdanov 1947, p. 257). This assertion laid the foundation for the following official academic research, due to which, many scholars gradually changed their academic positions. For instance, in the first edition of *Ancient Oriental History* (*Istoriya Drevnego Vostoka*) written by the Orientalist Avdiev Vsevolod Igorevich (1898–1978), Laozi was an enemy and a mystic (Myshinskij 2015, p. 345). However, in the 2nd and 3rd edition of *Ancient Oriental History* revised in 1953 and 1970, Laozi was thought to be a progressive thinker whose doctrine had elements of naive materialism and dialectics of nature, "Dao" is everywhere, and it is thanks to the existence of "Dao", that everything in the world could survive and thrive (Avdiev 1953, p. 670). Yang Xingshun's monograph *Ancient Chinese Philosopher Laozi and His Doctrine*[7] (*Drevnekitajskij filosof Lao-tszy i ego uchenie*) published in 1950 was also a positive response to Zhdanov's viewpoint. Yang wrote this book to expose the distortion of Laozi's theory by Kuomintang scholars and Western bourgeois scholars, and to prove that Laozi's philosophical theory was the embryo of Chinese materialism.

Yang Xingshun believed that Laozi's theory on "Dao" is a simple materialist philosophy, comparable to the ancient Greek philosopher Heraclitus. "Dao" does not have any artificial elements. It is the understanding of the natural world, the real world, and human life. It is not dominated by gods or divinities but follows a certain natural path (Dao). This Dao is inaccessible to our senses, it is the general law in philosophy, and it is independent of human will (Myshinskij 2015, p. 346).

In the 1960s and 1970s, the decrease in the cultural exchanges between China and the Soviet Union led to a drastic drop in the sinology research of the Soviet Union. During this period, Soviet scholars began to adopt Western academic approaches such as comparative literature, structuralism, and reception aesthetics to study literary and philosophical issues. They abandoned class interests and showed great respect for the diversity of different ideas and viewpoints. Questions and challenges began to emerge on the materialistic understanding of "Dao", and two camps of views appeared.

The first camp adhered to the theoretical vision and research method of materialism, and they regarded "Dao" as a scientific concept and a weapon against idealism and religious thought. For example, Pozdneeva Ljubov' Dmitrievna (1908–1974), whose views were a direct continuation of Petrov and Yang Xingshun, published *Atheists, Materialists, Dialectics in Ancient China* (*Ateisty, materialisty, dialektiki Drevnego Kitaya*) in 1967, believing that Zhuangzi's materialism and atheism were inherited from the materialism of Laozi and Liezi. She interpreted "Dao" as "nature, material, namely objective reality as opposed to human subjectivity" (Pozdneeva 1994, p. 389).

Another representative was Kozlovskij Boris Jur'evich (1898–1953), who further developed the arguments of Yang Xingshun and Pozdneeva. He thought that neither Daoist

studies should give way to bourgeois scholars, because they would distort it (from idealist point of view) or refuse to look for hints of materialism and idealism altogether, nor give way to Mao Zedong thinkers, because although they recognize the materialistic character of Daoism, they would place Laozi's materialism under Marx's materialism (Kozlovskij 1976, pp. 81–88). It can be seen that while insisting on materialist theoretical methods, Kozlovskij also recognized that the existing research in the Soviet Union had obvious traces of the Marxist theory of materialism.

The second camp mystified "Dao" and criticized the materialist interpretation of Laozi's philosophy, believing that Daoism had the feature of idealism, and the representative figure was Vasil'ev Leonid Sergeevich (1930–2016), who criticized the materialist interpretation of Daoism proposed by Yang Xingshun and Pozdneeva. He believed that there were indeed elements of materialism and dialectics in Laozi's theory, but more traces of idealism and mysticism could be found. Following the footsteps of French sinologist Henri Maspero, Vasil'ev also called Laozi "a melancholic mystic". He thought that "Dao" is a universal law of nature, the beginning and end of creation, and the foundation of profound metaphysics. Dao is everything and nothing, no one created Dao, but everything happens because of it and returns to it. No one can fully comprehend Dao. Our senses cannot touch it. What can be heard, seen, felt, and understood is not the Dao (Vasil'ev 1970, p. 229). Vasil'ev also compared "Dao" with the Indian religious concept "Brahman". Scholars believe that if substantive and metaphysical divisions are made in ancient Chinese philosophy, the transcendental "Dao" is similar to the transcendent "Brahma". If the above division is not made, Daoism is similar to the naturalism of pre-Socratics. Scholars believed that "Dao" is not created by man, and that all things originate from and return to "Dao". This view is consistent with Aristotle's understanding of "Nature". From the above-mentioned discussion, we can see that Vasil'ev's interpretation of "Dao" was similar to the "old-school" academician Alekseev. Vasil'ev was also the first to mention and praise Alekseev's work in the Soviet sinology literature.

Rubin Vitalij Aronovich (1923–1981), a researcher at the Oriental Institute of the Soviet Academy of Sciences, was another scholar whose views were contrary to that of Yang Xingshun and Pozneeva. He agreed with Alekseev and Vasil'ev's views on "Dao", saying that "Dao" is the universal rule and the mysterious origin of all things. It is the inner part of the world, it cannot be understood by emotion and reason, and it belongs to the category of mystical pantheism (Rubin 1970, p. 151).

The book by Vasil'ev and Rubin was a proof of a "thaw" in Soviet sinology, but it did not last long. In the 1960s, Sino-Soviet relations became sour, and Soviet sinology research once again fell into the hands of ideological control. The 1971 All-Soviet Scientific Conference of Sinologists became a turning point. In his report, the politician Senin Nikolaj Gerasimovich (1918–2001) reiterated Zhdanov's 1947 policy on the history of Chinese philosophy, and spoke highly of Petrov, Yang Xingshun, and Pozneeva, because their viewpoints were supported by the authorities. In contrast, Vasil'ev's viewpoint, which can be traced back to the academician Alekseev, was not recognized by the authorities. Senin sharply criticized Vasil'ev's book: "In this book, the author openly ignored the results generally recognized by Soviet researchers, and he questioned or even completely denied the existence of materialist ideas in China... Comparing Soviet sinologists with Western bourgeois sinologists in favor of the latter is something totally unacceptable to us" (Pozdneeva 1973, p. 157).

Lisevich Igor' Samojlovich (1932–2000), an expert in ancient Chinese literary theory, also challenged the view that Laozi's philosophy was materialism. He made a partial translation of the *Daodejing* with high quality and rich annotations. His views contradicted the prevailing one (Laozi and Zhuangzi are materialists), and he did not agree with the interpretation of "Dao" as "material", namely "a philosophical category that marks objective reality", which is similar to what Lenin wrote in his article "Materialism and Empirio-Criticism", because "the main feature of Dao is inaccessibility to human reason and emotion, while material is the perceptible objective reality" (Lisevich 1979, p. 10).

Sinology studies in the Soviet Period basically regarded Daoism as a theory opposed to Confucianism. At that time, Confucianism was criticized as a conservative and counterrevolutionary theory. The high-pressure political and cultural environment seriously undermined the independence and legitimacy of academic research. Many scholars had to interpret ancient Chinese philosophy in line with official ideology, and some even came to self-contradictory conclusions[8].

## 4. Post-Soviet Period—Contemporary Construction from a Diversified Perspective

After the 1980s, with the normalization of Sino-Soviet relations, Russian sinologists regained their enthusiasm for research. The collapse of the Soviet Union brought massive transformation to the mainstream ideology. With the evolution of Eastern and Western perspectives and the introduction of contemporary research methods, Russian philosophical research has undergone great changes in terms of the theoretical foundations, value orientations, and research methods, putting an end to the previous ideological manipulation. Researchers started to draw on diversified theories in their work. In 1983, the third issue of the periodical *Asian and African People* (*Narody Azii i Afriki*) published an article about the roundtable meeting on traditional Chinese culture, which can be deemed as the turning point. At the conference, scholars criticized the view of dismissing the development of Chinese philosophy as merely a historical fight between materialism and idealism. At the same time, there was a debate on whether the Chinese classical philosophy could be rationally explained.

For example, the early views of Feoktistov Vitalij Fyodorovich (1930–2005), a researcher on Chinese thought, were completely in line with the spirit of the 1971 All-Soviet Scientific Conference of Sinologists. In his article "On the Materialist Tendency in Xunzi's Philosophy" (*O materialisticheskikh tendentsiyakh v filosofskikh vzglyadakh Syun'-tszy*), he agreed with the views of those "recognized Soviet scholars" and regarded Laozi's theory as a naive materialist doctrine with elements of dialectic nature. After the 1983 roundtable meeting, he changed his view and pointed out: "If the Chinese philosophers did not come up with the same concepts of matter and consciousness as their European counterparts, then the attempt to classify Chinese philosophers as materialists and idealists using terms understood by Europeans is not convincing" (Feoktistov 1997, pp. 34–35).

Russian sinologists paid special attention to the translation and understanding of key philosophical words. Kobzev Artem Igorevich (1953–), a representative of Daoist studies in the post-Soviet period, divided the Russian sinologists of this period into three schools according to different research methods and viewpoints on Chinese classical philosophy. The first was the structuralists or logicists, whose methods can be traced back to the structuralist theory of Claude Levi-Strauss (1908–2009). Representative figures in this school include Spirin Vladimir Semyonovich (1929–2002), Karapet'jantc Artemij Mihajlovich (1943–2021) and Myall' Linnart Ehduardovich (1938–2010), a scholar of the Moscow—Tartu semiotic school. They believed that Chinese philosophy was a rationalist philosophical system composed of interrelated elements according to certain abstract rules. For example, Spirin was a groundbreaker who applied the method of modeling in sinology research. He advocated that Chinese philosophy did not exist in specific terms, but in a semiotic construction based on natural language, which can be interpreted in a rational manner. This means that the elements of this system (or category) should also be interpreted in a rational manner, and the system should be interpreted first before moving on to the elements. From the perspective of text structure analysis and mathematics, he translated the "Dao" of the *Daodejing* into a graph (grafik). A graph is very general and abstract in itself, but it can be interpreted as a great variety of concrete things. All of the concepts and ideas in the *Daodejing* is a point or multiple cross points in this graph, and "De" is a point on the graph. "Dao" and "De" are interpreted as mathematical terms by Spirin, who wrote "Mathematics has great methodological significance in the formation of Chinese philosophy, precisely because it brings the principles of rational argument" (Spirin 1976, pp. 212–19).

The second school is the interpreters or metaphorists, represented by Malyavin Vladimir Vyacheslavovich (1950–), Torchinov Evgenij Alekseevich (1956–2003), Grigor'eva Tat'yana Petrovna (1929–2014), and Zavadskaya Evgeniya Vladimirovna (1930–2002). They followed the traditional theories of Heidegger, Gadamer, and Ricoeur. In their view, it is impossible to explain Chinese classical philosophy with European philosophical terms because Chinese classical philosophy is a metaphor in essence, nor is it possible to explain it rationally at all. One can only "guess" the meaning of Chinese classical philosophy (Kobzev 1983, p. 65).

After the 1980s, the famous sinologist Torchinov began to study the historical religious methods of Daoism. He thought that the most important aspect of Daoist philosophy was natural science with cosmology as the core, believing that the "Dao" in the *Daodejing* had generative properties, and it was a unity of truth (ultimate meaning) and method (through path). This "Dao" was closely related to all things in nature and the practice of human life. Instead of being an abstract metaphysical "ontology" (the essence of the world), "Dao" was both "sensory and supersensory". He was convinced that there were two "Daos" (Double Dao) in the *Daodejing*, one was "Unnamable", which produced the universe, and the other was "Named", which produced myriad things (Torchinov 1994, p. 95). Generally speaking, Torchinov removed the influence of Stalin's "natural materialism"[9] when studying Laozi, and he no longer used simple "idealism" or "materialism" to characterize Laozi's thoughts, leaning toward the Marxist—Leninist "new materialism". However, there were also certain limitations. Torchinov did not thoroughly implement Mann's "materialism", that is, paying attention to social practice and revealing the meaning of life. Popovkin Andrej Vladimirovich (1974–) pointed out when evaluating the works of Malyavin and Torchinov that, in their works, they tried to understand Eastern teachings from the perspective of phenomenonological hermeneutics. Meanwhile, Malyavin and Torchinov also compared Laozi's theory with Russian intuitionism (Popovkin 2006, p. 164).

Malyavin, a famous contemporary sinologist, has made great contributions to the Russian study on Laozi. He published a series of works such as *Laozi—the Daodejing: a Book about the Way of life (Lao-Czy—Dao-De czin: kniga o Puti zhizni)* (2010), *Daoist canons in new translations by V.V. Malyavin (Daosskie kanony v novyh perevodah V.V.* Malyavina) (2017–2019), etc. He also translated *The Library of Chinese Classics—Laozi* from Chinese to Russian in 2009 with Li Yingnan 李英男, a Russian Professor at the Beijing Foreign Studies University. This translation combines the advantages of the translators from the two countries. They took Professor Chen Guyin 陳鼓應's *Annotation, Interpretation, and Comments on Laozi* as the parallel Chinese version, and paid great attention to the accuracy and elegance of the target language. Malyavin followed the modern hermeneutic theory of Heidegger-Gadamer. In his works, he regarded "Dao" as an independent concept with both objective and subjective dimensions, and the basis of matter and spirit co-exist in this concept (Malyavin 2010, pp. 698–99).

The third group are the symbolists, represented by Kobzev. He attempted to synthesize the above two methodologies. He believed that symbolism, or the so-called "symbol", was the essence of traditional Chinese philosophy, and that this symbolism required multifaceted, multidimensional (including metaphorically appropriate, scientifically concrete, and philosophically abstracted) interpretations. He clarified that the symbolic concept was characterized by both the feature of poetic language and the simplicity of mathematical formulas. The characteristic of traditional Chinese philosophical terms is that they can construct texts with metaphorical properties and rational interpretation.

On the basis of Kobzev's classification above, there are also some sinologists who have approached Daoism from a more international perspective, and are good at using research methods of Chinese and Western comparative philosophy. We think that this can be classified as the fourth school, that is, comparative philosophy, represented by Luk'yanov Anatolij Evgen'evich (1948–2021) and Grigor'eva Tat'yana Petrovna (1929–2014), etc. Luk'yanov was a representative figure in contemporary Russian Daoist research. In 1991, the People's Friendship University published Luk'yanov's monograph *Laozi:*

*Early Daoist Philosophy* (*Lao-tszy: Filosofiya rannego daosizma*). In 2008, Luk'yanov's latest translation of the *Daodejing* was published in Moscow. In order to express the hidden rhythm in prose language, the translation consisted of two parallel versions: the prose version and the poetic version, with the second version translated by his collaborator Abramenko Vladimir Petrovich (1932–2016). Luk'yanov believed that "Dao" was the symbol of the entire ancient Chinese philosophical culture. All natural and human systems in the world were adjusted according to the rhythm of "Dao", and they regenerated the material, spiritual, and ideal image of "Dao" in their own cycle. "Dao" was not so much a concept, but an organic whole that embodied a certain reality, forming the human–society– natural circle of the "Chinese universe". He attempted to explain the connotation of "Dao" with theories such as anthropology and cosmology. By combining relevant theories of Chinese and Western philosophy, he compared those primitive cosmological paradigms of "Dao" culture with the Indian culture of "*Aum*" and the Greek culture of "*Logos*". In 2020, Luk'yanov explained how to integrate Chinese "Dao" culture into Russian "*glagol*" culture[10] in his article "Prospects for Russian Translation of Ancient Books of 'Dao' Culture", and built a complete system of Russian national culture. In his view, Chinese spiritual culture evolved with the continuous interpretation of the keyword "Dao", which formed the inner circle of Chinese spiritual culture. It was also one of the eternal driving forces behind the development of Chinese society. It is a pity that Mr. Luk'yanov passed away from COVID-19 in 2021, which is undoubtedly a huge loss to the Russian study on Laozi. Another representative, Grigor'eva, published her book *Dao and Logos (Dao i Logos)* in 1992. The author compared Eastern and Western philosophy and believed that the relations among cultures boiled down to the unity in diversity. "Dao" and "logos" are very different in their origin and development, but they have similarities in knowledge orientation and philosophical meaning. In her book, she discussed the unique cultural paradigms of the ancient Chinese and ancient Greeks, explored the similarities and discrepancies of the basic concepts of philosophy between the two cultures, and finally, proposed the complementarity of Eastern and Western cultures and predicted that the two will inevitably meet and be compatible in the future (Grigor'eva 1992, p.41).

After the disintegration of the Soviet Union and the withdrawal of its mainstream ideology, Russian sinologists increasingly recognized the important role played by Chinese philosophy in promoting the development of world spiritual civilization, and paid particular attention to the modern relevance of traditional sinology and the prospect of the interaction between Chinese and Western cultures, making sinology research a broader interdisciplinary field. Modern and contemporary sinologists have carried out multi-level interdisciplinary research on Daoist literature from textual analysis, linguistics, semiotics, cultural studies, and even mathematics. This debate and synthesis of different research methods have been extended and continued in Russia today.

Entering the 21st century, the study on the *Daodejing* is thriving, and it has become a new trend in Russia to study the annotated Russian translations of the *Daodejing*. The monograph of Maslov Aleksej Aleksandrovich (1964–) *The Mystery of the Dao* (Misteriya Dao) created a new model for the study of the Russian translation of the *Daodejing*, and it is the first work that reproduces the tradition and feature of the annotation on the *Daodejing*. Not only did Maslov make his own annotations, but he also translated the full text of Wang Bi's 王弼 annotations into Russian for the first time. His interpretation was full of mysticism and religious rituals, and at the same time, he used symbolism to reveal the mysterious wisdom of ancient China vividly. He pointed out: "Since the prototype of Dao is the world, the world is symbolic. On the one hand, it is absolutely real, and everything does exist. On the other hand, behind this world, there is a more real, more valuable and more essential world. But this world is empty, hidden, invisible, illusory" (Maslov 1996, p. 76).

We mentioned the concept of "Double Dao" in the *Daodejing* previously, but Maslov disagreed with this point of view, arguing that although the contradictoriness of "Dao" left some room for the "Double Dao" interpretation, this might contradict the logic of the whole book, because "Dao" is everywhere in the *Daodejing*, while You 有—Wu 無

comes from the same origin but with different names ([Maslov 2005](), p. 118). That is to say, the concepts of You–Wu in Chinese philosophy complement each other, unlike the contradictory Being/Non-Being division in Western thought.

As Global Laozegetics and closer cultural exchanges between China and Russia thrived, the dissemination and influence of the *Daodejing* in Russia was no longer limited to academia, but was also felt by ordinary people. Malyavin once said that "Dao" is a kind of truth that should not be objectified in concept, rather, it should be made part of people's daily life experience. In addition to serious philosophical discussions, there are many writers who have studied and discussed the *Daodejing* in their own way. Take Burdonov Igor' Borisovich (1948–) as an example, where his translation is full of personal subjectivity and associations. His work of translation is named *Interpreting Daodejing in Lipovka* (Dao Deh Lipovka Vehj). In the first chapter, he uses his hometown of Lipovka to refer to "Dao", giving his hometown an abstract, perceivable yet indescribable meaning, and this was his personal rewriting and creation. In May 2018, the latest version of the *Daodejing* translated by Kondrashova Lyudmila Ivanovna was published. This edition added the calligraphy of the famous Chinese calligrapher Zhao Xueli趙學禮 and illustrations of the famous Russian artist Konyuhov Fyodor Filippovich. With both pictures and texts, this version has both literary and esthetic value, which promotes the cultural exchange between the Chinese and Russian people. When translating, Kondrashova bore the modern readers in mind, making Laozi's thoughts more popular and accessible to the public. During this period, scholars began to use scientific methods to conduct multi-angle, in-depth, and specific translation research on Daoism, trying to restore the *Daodejing* to the greatest extent possible in the Russian cultural context, and building a platform for Russian and Chinese people to have a cultural dialogue with each other.

Against the background of ever-growing passion in the translation and research on the *Daodejing*, in addition to the classic versions introduced above, many experts and scholars have tried to translate it in whole or in part. For example, Baranov Aleksandr Nikolaevich (1948–2021) published his translation of the *Daodejing* in 1998 based on R. B. Blakney's English translation, and he compared the *Daodejing* to the *Bible* and the *Bhagavad Gita*. In 1999, the Russian translation by Semenenko Ivan Ivanovich (1947–) was published in Moscow, which had precise language and detailed annotations. He pointed out that the *Daodejing* helped to reveal the mystery of life and to find oneself in the "Dao". In 2000, a translation by Polezhaeva YUliya was published, which adopted the translation strategy of domestication, thus losing the Chinese style to a certain extent. In 2002, Solov'eva Marina Pavlovna (1952–) published another version, which was more of a rewriting instead of a translation, since the author was trying to make the content easier to understand with his own narration. Vinogrodskij Bronislav Bronislavovich (1957–) has translated the *Daodejing* several times, and published a separate edition in 2014, which not only provided the author's latest version of translation, but also contained the author's unique interpretation. In addition, some translators tried to adopt a poetic style to translate the *Daodejing* such as Borushko Oleg Matveevich (1996), Kang Yu (1991), Feano (2001, 2005), and so on.

To sum up, Russian sinologists in this period completely removed the philosophical research model of the Soviet period, broke through the simple dichotomy of materialism and idealism, and freed themselves from treating the history of philosophical development as a response to class struggle, thus revealing the intrinsic characteristics of Chinese philosophy and culture. The diversification of theories has also promoted further innovations and explorations in Chinese philosophical and religious studies in Russia. It can be seen that Russian philosophical research on the "Dao" has been developed in a more multifaceted and open direction, and its connotation has become clearer and richer in the process of cultural exchange and collision. The Russian readers' understanding of "Dao" in the *Daodejing* has also been continuously enriched and improved.

## 5. Summary and Conclusions

The word "Dao" has a broad connotation and spiritual meaning that is "perceivable but indescribable". It is not only a microcosm of linguistic and cultural phenomena, but also a multi-faceted embodiment of historical philosophy. From the perspective of translation, with the deepening of the world's understanding of Chinese culture, sinologists have realized that "Dao" contains a wide range of meanings, and there is no Russian word that can cover all of the spirit contained in the cultural connotation of "Dao". In the period of Imperial Russia, sinologists had different understandings of "Dao", and they came up with different translations such as "Way" (*put'*), "God" (*bog*), "Reason" (*razum*) and so on. This kind of inconsistent translation not only affected the integrity of the terminology, but also led to the loss of cultural meaning of the source language. Nowadays, we tend to use the generally accepted transliteration of "Dao", which has penetrated into the Russian language and become a culture-loaded word with rich connotation.

A review of different understandings of "Dao" in the *Daodejing* revealed that Russian sinologists are greatly restricted and influenced by the theoretical background, academic prejudice, social, and political environment. They tend to be pragmatic in their research, which might be religiously and politically motivated. In terms of research methods and ideas, the Russian sinologists' research on "Dao" was affected by the political interference from the very beginning, and they were inclined to adhere to their thoughts or change their academic position with the changing ideology. In the research process and direction, the studies on Laozi in China and Russia were once in sync with each other. In terms of learning and reception, Russian scholars' understanding and translation of "Dao" has a strong national character. Russian sinologists are good at integrating the essence of traditional Chinese culture and the self-consciousness of their own nation, refining the connotations of "wisdom", "law", and "nature", which shows the Russian nation's insistence on sticking to its original culture and its expectation of traditional Chinese culture.

Most Russian sinologists highly appraise and respect the study of Laozi and "Dao". However, during the process of translation and research, they inevitably put "Dao" into the Russian cultural context and use Western philosophical or religious concepts to interpret Chinese keywords of traditional culture, compounded with the influence of social ideology and the limitation of the translator's identity, cultural distortion, and misreading may occur to a certain extent, which is also the research limitations from the perspective of others. In the Imperial Russian Period, researchers were influenced by the religious vision when studying Laozi. In the Soviet Period, one of the distinctive features was applying the theoretical vision and research methods of "materialism" in the studies of Laozi. Nowadays, the research on Laozi is increasingly diversified and multi-dimensional, and it can be seen that the differences in the research methods and theoretical perspectives in different periods have presented completely different understandings and even conflicts. This article takes "Dao", a keyword and core concept in the *Daodejing*, as the research object, investigated the understanding and interpretation of "Dao" in Russia in a historical review, and explored the interpretive trends and reception of the *Daodejing* in Russia, which, hopefully, will help to complement the traditional study on Laozi, and also enrich the depth and breadth of Global Laozegetics.

**Author Contributions:** Conceptualization, H.Z. and J.L.; methodology, H.Z.; software, H.Z.; validation, H.Z. and J.L.; formal analysis, H.Z.; investigation, H.Z.; resources, H.Z. and J.L.; data curation, H.Z.; writing—original draft preparation, H.Z.; writing—review and editing, J.L.; visualization, H.Z. and J.L.; supervision, J.L.; project administration, J.L.; funding acquisition, H.Z. All authors have read and agreed to the published version of the manuscript.

**Funding:** The Project Supported by National Social Science Foundation of China (Grant No. 20CZW031).

**Institutional Review Board Statement:** Not applicable.

**Informed Consent Statement:** Not applicable.

**Data Availability Statement:** Not applicable.

**Conflicts of Interest:** The authors declare no conflict of interest.

## Notes

[1]　Sinology, in a narrow sense, refers to a comprehensive interdisciplinary study related to China and Chinese culture, it mainly involves language, literature, history, religion, and philosophy. It also includes Manchus study, Mongolian study, Tibetan study, Tangutology, and Khitan study in a broad sense, but sinology in this article is used in a narrow sense.

[2]　Target-culture-oriented translation strategy refers to the translation strategy in which a transparent and fluent style is adopted to minimize the strangeness of the foreign text for the target language reader. It means making the translated text recognizable and familiar and thus bringing the foreign culture closer to the reader in the target culture. It is just like Friedrich Schleiermacher's standpoint, which "leaves the reader in peace, as much as possible, and moves the author towards him".

[3]　The manuscript is kept at the Archive of foreign policy of the Russian Empire, f. SPb. Main archive 1–5, 1817–1840; d.1, folder 2, l.36, autograph.

[4]　Tolstoyism arose in the Russian Empire in the 1880s on the basis of the teachings of Leo Tolstoj. The main principles are: "non-resistance to evil by violence", "moral self-improvement", "renunciation of hostility with any people ('love your enemies')". "Non-resistance to evil by violence" was put forward in response to the reality of Russian and Western capitalist society at that time; "moral self-improvement" focused on exploring human nature, thus eliminating the root cause of evil; "renunciation of hostility with any people ('love your enemies')" was a beautiful vision that underpinned the future of human society. Tolstoj's interpretation of "Dao" was the basis of the third principle.

[5]　By Orientalism, I mean what defined by Edward Said as "a style of thought based upon an ontological and epistemological distinction made between 'the Orient' and (most of the time) 'the Occident,' and a Western style for dominating, restructuring, and having authority over the Orient."(Edward 1979, pp. 2–3) David Schimmelpenninck Van Der Oye and Susanna Lim both showed that Orientalism in Imperial Russia and the Soviet Union is more complex than the Saidian one.

[6]　Slavism is usually regarded as one of the main trends in Russian religious philosophy in the middle of the 19th century. The philosophy of history and the Russian view of history are its main subjects of study. In the philosophical realm, Slavism values the role played by faith. Faith is not understood as rational intuition or supersensory perception, but rather as the principle that unites the elements and forces of human intellect such as will, sensibility, and understanding. Only "faith" can overcome the limitations of individual rationality.

[7]　The monograph *Ancient Chinese Philosopher Laozi and His Doctrine* summarizes the position of Laozi's thoughts in the history of ancient Chinese philosophy, and gives an account of the social and historical situation as well as the ethical theory when the *Daodejing* came out. The book also discusses the materialistic nature of "Dao", and how the Western European bourgeoisi think of the *Daodejing*. In addition, it also analyzes the research on the *Daodejing* in Russia before the revolution, and it attaches the preface and translation of the *Daodejing*. This work had great influence at that time and was the representative work of studying the *Daodejing* in the Soviet era. It was translated into Chinese in 1957, and the Chinese title is ≪中國古代哲學家老子及其學説≫.

[8]　During the same period, there were also academic debates about the nature of Laozi's philosophy in China. For example, Feng Youlan 馮友蘭, Ren Jiyu 任繼愈, etc. believed that Laozi was a materialist, while Hu Ruichang 胡瑞昌, Hu Ruixiang 胡瑞祥, Yang Liuqiao 杨柳橋, etc. argued that Laozi's philosophy could only be objective idealism.

[9]　This article distinguishes between two kinds of "materialism": Stalin-style "materialism" and Mann's "materialism". The establishment of the "materialism" of Marxism in a broad sense is based on Marx and Engels, while the "materialism" of Mann has been widely spread in the Soviet Union because of the promotion of Plekhanov and Lenin. However, Stalin transformed the Marxist–Leninist "new materialism" into a narrow "natural materialism". Stalin-style "materialism" has always been the dominant research paradigm in Soviet social science research, and the views of Yang Xingshun we mentioned above are closer to Stalin's "natural materialism".

[10]　Luk'yanov regarded "glagol" (Глагол) culture as the archetype of Russian culture from the perspective of philosophy and culture, believing that the Russian "glagol" was the same as the Chinese Dao, the Indian Aum, and the Greek Logos, and they all had the meaning of words, speech, and discourse.

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
