# Peer review of "Interpretive Trends and the Conceptual Construction of the Daodejing’s Dao in Russian Sinology: A Historical Overview"

_religions, doi:10.3390/rel13090825_

Round 1

Reviewer 1 Report

I have missed the contextualization of this research in the leading and long-established Western sinology and Daology (in the introduction and especially in the conclusions, and concerning the XXIst century Daology in particular) - for example, whether/how the interpretations of Dao by Russian daologists are related to some Western (especially French, English) investigations. There are only some short remarks about this.

Although the research design is quite clear (historical overview of three periods), but its methodology is not described at all (only in the conclusions one can read, that there was a "review"). The article lacks more critical analysis and the position of author/authors, as well as analytical or hypothetical questions.

Not all arguments and discussion seem coherent, balanced and compelling.  For example, it is not clear, how the author/ors treat the word "sinology" - as an academic discipline or any interest in China? This is especially the case with L.Tolstoy's, who was the Russian writer, interpretation of Dao and Daoism - why he was given far more attention (almost one page), while Vasil'ev Vasilij Pavlovich, who according to V.M.Aleksejev, was the founder of the first Russian school of sinology, was given only one short paragraph? It would be useful for the authors to include some information of Aleksejev's article (from the book "Nauka o Vostoke", 1982, p.160-185) about the first and second Russian schools of sinology, in order to provide a broader academic context of the interpretation of Dao  and Laozi, especially since in Alekseev's view, only the second school have "rehabilitated" Laozi's book (Daodejing) as Chinese classics, while the first one (especially Vasil'ev himself) was interested more in Buddhism. I have also missed the relation of Russian Daology with its Buddhist studies, which were maybe far more active and progressive than in the Western sinology in the beginning of XXth century. I have also missed the discussion of one of the latest and biggest Russian academic works in Daology - a new improved translation of Daodejing by V.V.Maliavin with his own analysis, comments and translated comments by Chinese commentators (Daodejing Laozi. Kniga o puti zizni, Moskva, 2010), because he made a huge contribution to the Russian Daology. He was only mentioned in this article as a representative of one of Russian "post-Soviet schools" of sinology, as divided by A.Kobzev. However, even inclusion of Kobzev's article and classification of schools into this article could be related more to its topics, namely, the discussion of the interpretations of Dao by their representatives (especially  by Zavadskaya, Maliavin, Kobzev himself). I have missed even the mention of important book by T.P.Grigor'eva "Dao i Logos" (1992), the translation of Daodejing by Vinogrodskij (in Antologiya daoskoj filosofii, 1994). I have missed more critical and analytical view of Kobzev's classification of Russian schools of sinology- for example, to which school they would classify Luk'yanov, Semenenko (see for example his book Laozi. Obresti sebia v dao, 1999) and Maslov? What about the comparative research (the research on Dao and Logos by Grigorjeva, mentioned above, and others) - should it be classified as a new school or not worth of any discussion?

I have also some doubts of this tripartite historical periodization of Russian daology (sinology), since many sinologists, discussed in the article, were working in both, Soviet and post-Soviet periods. Finally, the article lacks the discussion of a young or new-generation Russian sinologists (that is, born in 1970-1990).

Some expressions, found in the article, should be more clarified (for example, "their research was mystical" (p.1), "the researchers of this period tended to be more objective and rational" (p.2), "Slavism tendencies" (p.5).

Some conclusions of the article are not related enough to the investigation - for example, the statement "Russian scholar's understanding and translation of "Dao" had a strong national character" and relation to "Russian cultural context" (p.11-12) - but there was no extensive discussion about this character and cultural context in the investigative part. It was mainly concentrated on the ideology of materialism/idealism.

I have some doubts of inclusion of "Global Laozigetics" into the list of article's e keywords, since it is only mentioned in the last sentence. There are far more important and more used words here, such as "materialism" and "idealism" (which also should be used in this article more critically and carefully).

Although I don't feel qualified to judge about English style and language, but have few remarks: first, the word "Daoism" should be written everywhere the same - now there are few cases with "taoism" (p. 5,7). Second, there is established English translation of Dao as "the Way" instead of the "road" (as it is written in this article, in the conclusions).

Reviewer 2 Report

This is a highly original and informative paper, a valuable contribution to the field, but also a good, engaging read for the non-expert, as well. There are, however, some issues which, once addressed, would make the paper even more readable and clear. Overall, the paper needs extensive editing of English language and style. (L: Line, Ls: Lines):

A. Language- and sentence structure-related issues

1. There are problematic word choices

L8: reflects the missing and compensation of cultural connotation

L16: acceptance instead of reception

L234: materialized understanding

L94: publicly published

2. There is inconsistency in some terms and names

Daodejing or the Daodejing? (e.g. Lines 4, 14)

L224: Taoism or Daoism?

Ls182 and 309: Heracleitus-Heraclitus

3. There are grammatical oversights. The author should edit thoroughly. Some examples:

L41: have been and is closely related

L554-555: in their research, which are

4. In many instances sentences are too long and need to be separated

E.g. Ls124-125, 302-303

5. Informal language  

L523: Take .... for instance

L557: sinologists tend to stick with their thoughts  

B. Content- and overall structure-related issues

1. Ls16-20: The last sentence of the Abstract describes an endeavor not actually taken up in the paper: underlying reasons of cultural filtering and misreading. It is advisable that the author limits the scope of the article to what is stated immediately after: to "outline the century travel of Dao in Russia" (even though 'century travel' is not a felicitous phrase).

2. Some terms could be elaborated on in the text or in footnotes:

L66: target-culture-oriented translation strategy

Is this a term in the literature or does it belong to the author? Some clarification is needed.

L154: A footnote on Tolstoyism would be helpful; Tolstoy’s encounter with Daoism is a highly interesting part of the paper.  

Ls210-211: A further note on this point

3. The concluding sentences of some paragraphs are elliptical, making bold or important claims without adequate justification or attention to nuance.

Ls527-528: He blurred the core concept in his translation and this was purely personal rewriting and creation.

Ls390-392: Despite this, a series of valuable academic results were achieved on the research of Daoism and ancient Chinese philosophy by the Soviet Sinology circle. Some elaboration here.

L233: Slavism tendencies. A footnote on Slavism here would be helpful.

L210-211: his intention to justify Western colonial rule cannot be ruled out.

C. Other issues

Ls210-211, 14-16: repetition

L85: the author should state whose the translation of the Russian translations is in a footnote

L573: Openness and compatibility

This is not clear. Compatibility with what?

Ls574-578: The last sentence of the conclusion reads like a eulogy or “manifesto.” A more sober defense of the value or contribution of this type of research would be more appropriate and convincing.  

Finally, I would suggest an alternative title for the paper:

Interpretive Trends and the Conceptual Construction of the Daodejing’s Dao in Russian Sinology: A Historical Overview
